# Bioburden Variation of Filtering Face Piece Respirators over Time: A Preliminary Study

**DOI:** 10.3390/ma15248790

**Published:** 2022-12-09

**Authors:** Vittorio Checchi, Marco Montevecchi, Leoluca Valeriani, Luigi Checchi

**Affiliations:** 1Unit of Dentistry and Oral-Maxillo-Facial Surgery, Department of Surgery, Medicine, Dentistry and Morphological Sciences, University of Modena and Reggio Emilia, 41125 Modena, Italy; 2Unit of Dentistry, Department of Biomedical and Neuromotor Sciences, University of Bologna, 40100 Bologna, Italy

**Keywords:** respirator, Filtering Face Piece, bioburden

## Abstract

Background: The microbial contamination of a respirator can be evaluated through a count of the number of bacteria living on a non-sterilized surface (bioburden). This preliminary study investigated the external contamination of two different FFP2s over time by studying the bioburden values in increasing exposure times. Methods: FFP2 respirators of two different brands were used during routine clinical settings and examined through the bioburden test; for each brand, three devices were tested at 8, 16, and 30 h. Results: No significant differences were observed between mask brands (*p* = 0.113). There were only significant CFU differences between each mask and its control (*p* = 0.027 and *p* = 0.004). Conclusions: Both brands of respirators were found to be contaminated and this contamination increased with the increase in exposure time. Further studies are needed to investigate the exact amount of contamination that could be considered acceptable before discarding each used mask.

## 1. Introduction

According to EN 149: 2001 European standards, Filtering Face Pieces (FFPs) are categorized into three different sub-classes based on their efficiency in aerosol filtration towards powders having granulometry of 0.02–2 µ. FFP1 shows a total filtering efficiency of at least 80%, FFP2 presents a minimum total filtering efficiency of 94%, and FFP3 has a minimum total filtering efficiency of 99% [1,2].

A recent publication has shown that the filtering capacity of FFP2 masks remains almost unchanged even after many hours (40 h) of continuous use in a periodontal setting [3]. Although the deduction is the result of a preliminary study, it suggests the possibility of the prolonged use of these devices with clear environmental and economic benefits. If these results were confirmed, they would bring with them relevant aspects worthy of further investigation.

Due to the fact that an effective method of disinfecting respirators while maintaining the filtering capacity and/or the structure of the mask has not yet been found [4,5], the World Health Organization and the national Centers for Disease Control of many countries have recommended appropriately extending the use time and frequency of single-use respirators [6,7]. 

Several scientific articles have been published in the literature that reveal how in recent years, probably following the pandemic, the practice of reusing respirators has become very common and practiced [8]. The extended use and reuse of respirators and masks are very common all over, especially during pandemics and outbreaks and when supplies are short and demand is high [9,10]. 

Among these aspects, it would be essential to understand what contamination the device undergoes during its use. Contamination results from two main factors: the environment and the operator. Both are noteworthy, but it is logical to hypothesize a risk of greater importance for the wearer of the device due to microbiological contaminants of environmental origin. If well-handled and managed during use and storage, the contamination of FFP2 from the environment can be identified from its exposure to aerosol and spatters [11,12].

Regarding spatter protection, the regular use of face shields is considered essential to limit the face masks’ external contamination [13,14]. These devices are a very important personal protection that has long been proven to be concretely capable of protecting the operator’s face from splashes [2,15].

Assuming that a rightly sized face shield is regularly used and that the oral respirator is properly managed, it can be assumed that the variables constituting the finished product (shape, material, seams, etc.) may affect the tendency of the FFP2 mask toward external contamination.

Bacterial contamination of an object is generally described by the term bioburden, and is normally defined as the number of bacteria living on a surface that has not been sterilized [16].

The term bioburden is most often used in the context of microbial limit testing, which is performed on pharmaceutical products and medical products for quality control purposes [17,18]. Products or components used in these contexts require the control of microbial levels during processing and handling. Microbial limit or bioburden testing on these products proves that these requirements have been met [19]. The unit of measure for this test is generally the Colony-Forming Unit (CFU), and European Law defines a limit for the respirator marketing of 30 CFU/g (UNI EN 14683:2019).

The microbial contamination of a complex device such as a respirator must be carefully considered. Depending on the specific interest behind the examination, it can be divided into different areas: the external, the intermediate, and the inner layer [3]. In addition, the characteristic porosity is another relevant aspect that influences microbial contamination.

The aim of this preliminary study was to investigate in vivo and over time the external contamination of FFP2s. By studying microbial contamination using a specific test, two similar models of FFP2 respirators (different brands) were investigated with increasing exposure times in a clinical periodontal setting. This preliminary study was carried out to test the effect of mask type and wearing time on bioburden. Data obtained will be used in a larger further study.

## 2. Materials and Methods

FFP2 respirators of two different brands were examined, and for each brand, three devices were tested at 8, 16, and 30 h. For each brand, one control was tested. Clean AIR 20300 (Eleco S.r.l., Bergamo, Italy) is a no-valve FFP2 model with a horizontal fold and external layers of polypropylene, a naturally hydrophobic material (Figure 1).

Crosstex Isolator N95 (HuFriedyGroup, Chicago, IL, USA) is a no-valve model as well, flat-folded with a duckbill design, made of 100%polypropylene (Figure 2). 

Both models are marked and recommended as non-reusable (NR), meaning that they can be used for only one single shift (according to EN 149:2001 + A1:2009). Each piece was packed individually in hygienic wrapping to prevent contamination before use.

For each FFP2 model, three samples were worn by the same operator (LC) in a periodontal private practice and held in place continuously during clinical activity and between each patient, until reaching a definite time. The operator was familiar with both models and a qualitative fit test had been performed and passed for both prior to this study (Louis M. Gerson Co. Inc., Middleborough, MA, USA). Clinical procedures, based mainly on surgical and non-surgical periodontal therapy, included procedures that involve the use of high-speed handpieces and ultrasonic devices, with the consequent generation of a high amount of droplets and aerosol [20]. No air polishing was ever applied. For the rotating instruments as well as for the ultrasonic devices, dental unit water was used for cooling. The evacuation was established by means of one conventional dental suction with a cannula of 3.0 mm in diameter. All clinical activity was conducted in two operatory rooms with two continuously open bottom-hung windows, and after each working session, both windows were kept completely open for at least 15 min. Before each treatment, all patients were disinfected using chlorhexidine gel for 2 min in the oral cavity [21]. The operator always wore a protective shield (Univet, Rezzato-BS, Italy) over his face and did not use a surgical mask over the respirator (Figure 1 and Figure 2). When the definite time was longer than the normal daily working hours, the respirator was carefully placed into its original wrapping and kept in a closed container until the next day, to completely cover the defined time. The respirator was always handled with sterile gloves and direct contact between the operator and the respirators was avoided. Once the correct time was reached, each respirator was then individually sealed in a clean plastic envelope for lab evaluation. Six FFP2 respirators (three for each typology) were not clinically used to act as a control.

### 2.1. Bioburden Test (BT)

The biological load was determined by sampling the core region of the outer surface of the respirator. The test was made with respect to the European Law UNI EN ISO 11737-1:2018. A Tryptic Soy Agar (TSA) contact plate and a Sabouraud Dextrose Agar (SDA) contact plate were placed in contact with the external surface of the respirator for ten seconds by applying light pressure. The operation was carried out in a sterile environment under a laminar flow hood.

Given the porosity, and therefore the permeability of the material, in order to specifically focus on the microbial load of the external part, this sampling method was chosen as an alternative to more common methods using paddle blenders and immersion.

Subsequently, the test proceeded with the incubation of the TSA at 35 degrees for three days to count the bacterial colonies formed on the plate, and with the incubation of the SDA plates for five days at 24 degrees for the detection of molds. Both incubations took place under aerobic conditions. The count was made by the naked eye to quantify the number of colonies grown on the agar. The two counts deriving from TSA and SDA contact plates were then added together to determine the total charge (bioburden) of the CFU/respirator (CFU/resp).

### 2.2. Statistical Analysis

Descriptive statistics synthesized the raw data (mean and standard deviations). A linear mixed-effect model was used to evaluate the effect of brand and time (fixed effects) on bioburden; an intercept was used as a random effect. ANOVA univariate analysis was used to compare the brands and, after verifying the non omoschedasticity of variance by means of the Levene test, the Tamhane test was applied for multiple comparisons. α-level was a priori set at 0.05.

## 3. Results

The experimental setting is reported in Table 1.

No significant differences were observed for the mask brands (ANOVA F-test: 2.472, *p* = 0.113). The Tamhane test evidenced a significant CFU difference only between the Clean AIR 20300 and the control (*p* = 0.027) and the Crosstex Isolator N95 and the control (*p* = 0.004).

Table 2 describes the effect of time on the bioburden of each brand. The bioburden increased with increasing time for both brands.

The linear mixed effect model confirms the not significant effect of the brand on the bioburden (*p* = 0.08) and suggests the significant effect of time (*p* = 0.001). In comparison with 30 h, the estimated bioburden is equal to −9 CFU at 8 h and −5 at 16 h, denoting an increase in bioburden when shifting from 8 to 30 h (Table 3).

## 4. Discussion

Respirators are designed as inhalational protection devices and are defined as Personal Protection Equipment (PPE) for their marked protective benefits [22,23]. Both the Occupational Health and Safety Administration (OSHA) and the National Institute for Occupational Safety and Health (NIOSH) define PPE as “the last line of defense”, encouraging scrupulous controls to reduce environmental exposure [24].

In Europe, filtering facepiece respirators must present the features indicated by the EN 149:2001 (+ A1: 2009) standard, which states that these masks must have specific characteristics of breathability, accumulation of CO_2_, inward leakage, and flammability. The EN 149:2001 (+ A1: 2009) standard also requires that the filtering capacity of the masks are tested with both NaCl aerosol particles with a median diameter distribution between 0.06 and 0.10 μm and with paraffin oil aerosol particles with a median diameter distribution between 0.29 and 0.45 μm. It should be emphasized that no bacterial filtration efficiency test is required [25].

In the USA, respirators must comply by law with the N100 or NIOSH N95 standard (42 CFR Part 84—Approval of Respiratory Protective Devices, United States Government Publishing Office, https://www.ecfr.gov/, accessed on 1 October 2022). N95 respirators are tested for resistance to an NaCl aerosol with a median particle distribution of 0.075 ± 0.020 μm and must have a filtration efficiency of at least 94%. Even in this case, like European respirator facepieces, they are not tested for bacterial filtration efficiency [25].

While several studies have highlighted the relevance of the use of respirators or face masks against the transmissions of respiratory viruses [26,27,28], the accumulation of pathogens on the masks due to human saliva, nebulized oral biofilm, and exhaled breath represents a possible underestimated biosafety concern.

The recent SARS-CoV-2 pandemic has strongly influenced the use of oral respirators in the medical and dental field, while also raising new questions. The initial lack of availability of these protective devices has stimulated their prolonged use if not “reuse” [29]. An interesting publication in 2021 [3] showed how the filtering capacity of FFP2 masks can remain almost unaltered for many working hours (up to 40 h). This result supports its prolonged use, assuming, however, that it is associated with a series of procedures aimed at reducing contamination of the device [2,13].

Multiple factors are involved in the bioburdens of masks, such as face cleanness, mask types, and speaking. Based on the results of a recent study, mask type is a critical factor that has a direct relationship with bioburden values; in fact, masks with low airflow resistance and high filtering efficiency are recommended. Moreover, washing the face could reduce the bioburdens on the face, but not the mask, and speaking could increase the mask bioburdens [30].

In our opinion, a particular distinction should be always placed between contamination resulting from the individual wearing the device and that resulting from the external environment. The latter is in fact the most dangerous for the individual as it is potentially loaded with new pathogens.

Among the possible and most common precautions suggested for the reduction in the bioburden of the external portion of an FFP2, the use of a protective screen is undoubtedly included [2,15]. Therefore, the present research aimed to investigate the level of contamination of the external surface of FFP2 when regularly used with protective screens.

The two models were chosen for two main reasons. The first one was their wide diffusion and commercial availability in our country at the study time. The other one is related to their macroscopic resemblance and specific peculiarities that distinguish them from each other. The Clean AIR 20300 is characterized by a softer inner portion, the presence of polyurethane foam on the nose area, and some folding lines on the front edge. 

The evidence of this study shows that despite the use of protective screens and specific management precautions (handling them with specific caution), both FFP2s tested incurred a real and increasing external contamination. However, it is difficult to demonstrate the risk level of such contamination. Reference data are lacking, but undoubtedly the contamination occurs in an increasing manner and is independent of the type of device tested.

The results of this preliminary study could lead to the conclusion that both tested masks present an increasing trend in bioburden with increasing exposure time. No similar studies have been previously performed on FFPs, but there are data available regarding surgical masks [10,31]. Unfortunately, however, these tests were performed in consideration of the total bioburden (internal and external, together) and therefore the results are not comparable.

A recent investigation aimed to examine the microbiological contamination of surgical masks during dental procedures. All masks used during treatment displayed bacterial contamination, indicating that they have been contaminated by aerosol-producing dental treatments. The authors also concluded that used masks have the potential to be a source of bacterial contamination of the hands [31].

Even if the study by Gund et al. is the one that most resembles the present manuscript in terms of objectives and type of investigation, the numerous differences between these two studies make comparison difficult [31]. First of all, Gund et al. evaluated surgical masks and not FFP2 respirators, and the operator wore only glasses as PPE and not a face shield, which has a higher coverage area of the face. Other substantial differences can be summarized in the type of procedure (several procedures vs. a single repeated procedure) and in the time of use (30 min vs. hours). Consequently, the final results must also be interpreted with care since Gund’s study indicates the presence of a generic <100 CFU [31], a range too wide to be compared to the low values found in the present manuscript.

Surgical masks are known to be a good substrate for microbial growth, holding moisture very well [32,33], and a potential source of bacterial shedding that could increase the risk of site infections. Moreover, it has been shown that the bacterial count on surgeon’s masks is directly proportional to the operating time, i.e., bioburden increases with prolonged wearing time [34]. Prior studies have proposed prolonged wearing time, speaking, and poor facial hygiene among the factors contributing to an increase in mask bioburden [30,34].

Therefore, it could be hypothesized that this bacterial load will be even greater after the canonical 8 working hours a day. For instance, the Belgian government recommended that after 8 h of regular use or 4 h of intensive use, the face mask should be replaced [33].

Our preliminary study showed how bioburden values present significant differences between tested and control masks, and that there is no statistically significant difference between the bioburden of the two different kinds of FFP.

These data, confirmed by the existing literature, confirm the fact that the masks are contaminated externally, regardless of the type, shape, and materials of which they are made, and that this contamination increases with increasing time of use.

Considering that both models weigh around 9 g, the external contamination observed after 30 h of continuous use remained under the threshold value of 30 CFU/g indicated by the European Law for the respirator’s production and commercialization (UNI EN 14683:2019). From this consideration, it is possible to speculate that, in this clinical setting, both respirators seem to undergo limited external contamination even after a long period of use.

One of the major limits of this study, as with other studies that have evaluated bioburden [31], is that the rubbing of the mask on the agar plate may not have made all the bacteria adhere, i.e., they could have crept into the pores of the respirator. This aspect may underestimate the results of the study. 

The key point to clarify, therefore, is what degree of bioburden is considered acceptable and compatible with the use of a mask. In other words, it would be important to be able to define whether a certain contamination value is considered high or low and, ultimately, tolerable and accepted.

The scientific literature does not help us on this specific issue and the question appears to be unresolved.

Only after having clarified this aspect, it could become reasonable to investigate whether particular attention or specific procedures (such as the use of a surgical mask over the FFP2 or mask renewal between one patient and another) could constitute a concrete advantage in ensuring a longer use of the device.

## 5. Conclusions

This preliminary study investigated the external contamination of two different FFP2s over time by studying bioburden values in increasing exposure times during a routine clinical setting. The two masks are quite similar in shape and constituent material, and, therefore, the absence of statistically significant differences between them could be expected. Both were found to be contaminated and this contamination increased with the increase in exposure time. Further studies are needed to investigate the exact amount of contamination that could be considered acceptable and tolerated before discarding each used mask.

## Figures and Tables

**Figure 1 materials-15-08790-f001:**
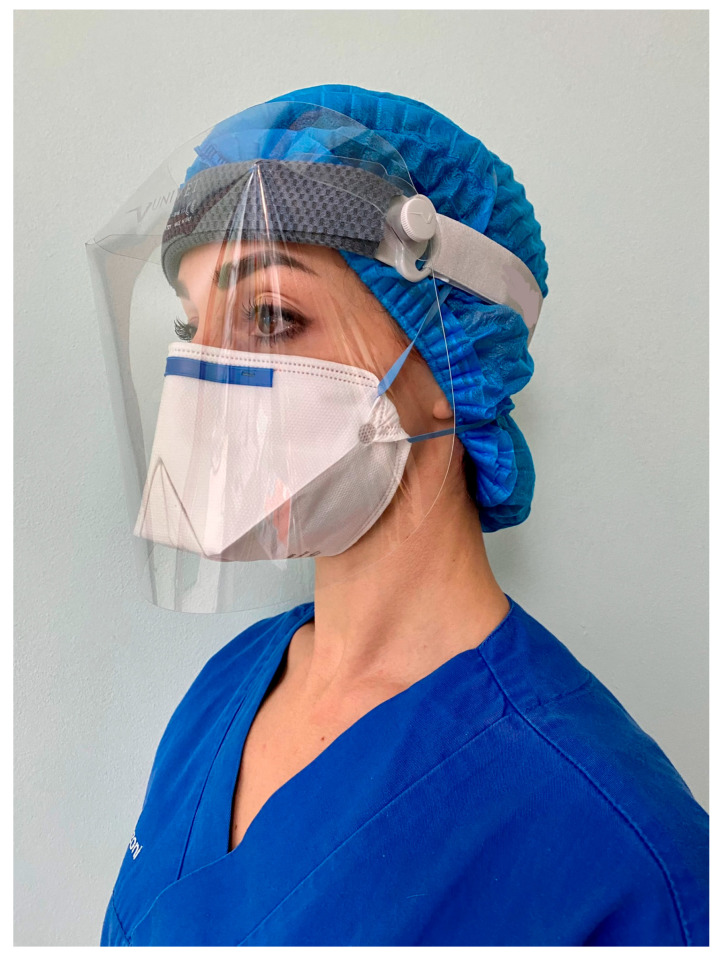
Tested Clean AIR respirator, worn during clinical procedures under a protective shield.

**Figure 2 materials-15-08790-f002:**
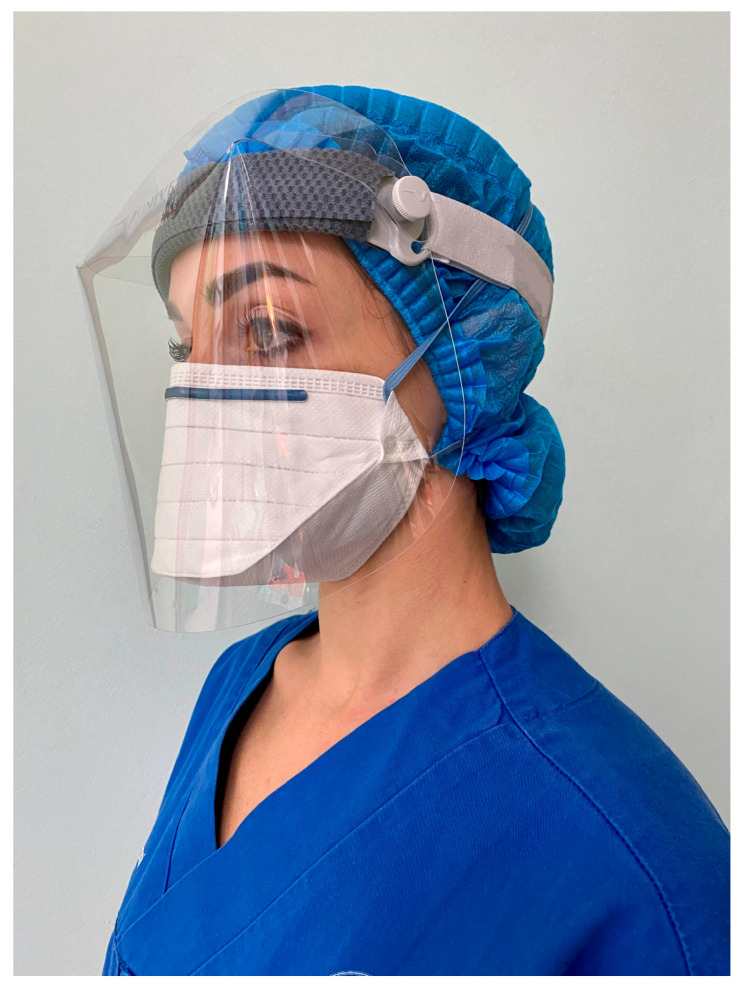
Tested Crosstex respirator, worn during clinical procedures under a protective shield.

**Table 1 materials-15-08790-t001:** Bioburden raw data (CFU) by type of protective device, time, and experimental setting.

	Device	Time (Hours)		
		8	16	30
Clean AIR 20300	First	2	8	14
	Second	2	8	13
	Third	2	7	14
Crosstex Isolator N95	First	4	6	10
	Second	4	6	11
	Third	3	6	10
Clean AIR 20300 (Control)	First	<2	<2	<2
	Second			
	Third			
Crosstex Isolator N95 (Control)	First	2	2	2
	Second			
	Third			

**Table 2 materials-15-08790-t002:** The mean values (standard deviation) of bioburden for each mask type across time.

Mask Brand	8 h	16 h	30 h
Clean AIR 20300 (*n* = 3)	2.00 (0)	8.00 (0.58)	14.00 (0.58)
Crosstex Isolator N95 (*n* = 3)	4.00 (0.58)	6.00 (0)	10.00 (0.58)
Control	2.00	2	2

**Table 3 materials-15-08790-t003:** The linear mixed effect model estimates of bioburden (CFU) variation.

Parameter	Estimate	Standard Error	*p*	95% Confidence Interval	
				Lower Limit	Higher Limit
Intercept	11.444	0.592	<0.001	10.174	12.715
Brand *	1.111	0.592	0.082	−0.160	2.382
8 h	−9.167	0.726	<0.001	−10.723	−7.610
Time **					
16 h	−5.167	0.726	<0.001	−6.723	−3.610

* Crosstex Isolator N95 as reference. ** 30 h as reference.

## Data Availability

Not applicable.

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
