# Peer review of "Bioburden Variation of Filtering Face Piece Respirators over Time: A Preliminary Study"

_materials, 2022, doi:10.3390/ma15248790_

Round 1

Reviewer 1 Report

The experimental design involved only two brands and three devices. What is the basis for the selection of the devices? For pilot scale study, this small number of sampling is not enough. 

The test samples taken were from the non-reusable category. What is the purpose of studying the non-reusable items? 

Is there any antibacterial or antimicrobial resistance tests conducted on the  protective face shields before the real time study with the patient? What is the porosity features of the test masks? What is the age group of the patient? 

Possible microbial loading is the effect primarily due to the surface properties? Is the surface hydrophilic or hydrophobic? 

This paper does not given any photographic images of the Face shielding masks. It is nice if any representative images are given.

The microstructure of the face shielding masks is necessary. More scientific analysis is required to understand the kind of contaminations. The data present in the study is not sufficient. 

Author Response

- The experimental design involved only two brands and three devices. What is the basis for the selection of the devices? For pilot scale study, this small number of sampling is not enough.

Dear Reviewer, thank you for your comment. Given the difficulty in finding valid PPEs during the pandemic period, the two brands of masks selected for the study are the FFP2 masks which were used in our clinical routine in that period. The two brands display macroscopic similarities but also specific peculiarities that distinguish them from each other. The Clean AIR one is characterized by a softer inner portion, the presence of a polyurethane foam on the nose area, and some folding lines on the front edge. All aspects that make this model more comfortable for our perception. This description has been reported on the text (Lines 204-208). In conclusion, the selection of the two brands was consequently moved by the product availability at the study time and on the intention to compare two very similar products.

Since there were two brands and we needed to test different times of use, we decided to use 3 devices for each group. A pilot study is a small-scale preliminary study conducted to evaluate feasibility, duration, cost, adverse events, and improve upon the study design prior to performance of a full-scale research project. One aspect of pilot and feasibility/preliminary studies that remains unclear is the required sample size. There is no consensus but only recommendations. Moreover, for mixed models that have become the most popular method for analyzing repeated measures, validated power and sample size methods exist only for a limited class of mixed models [Muller K, Stewart P: Sample size for linear mixed models. Linear model theory: Univariate, multivariate and mixed models. 2006, New York, NY: Wiley]. In addition, most of these methods are based on approximations, and make simple assumptions about the study design. No previous studies have compared the bioburden of these two types of respirators across time; consequently, no variance or correlation pattern among repeated measures exist. This paper represents a small-scale explorative feasibility study suggesting low intra time variability among the devices of each brand and stable correlation pattern among time of each brand. These aspects, more than comparative results, may be used for implementation of a pilot trial of adequate power.

Due to the small number of sampling and as a consequence of your pertinent consideration we have decided to rename the manuscript from “pilot study” to “preliminary study” (Lines 3, 12, 32, 75, 78, 215, 246 and 273).

- The test samples taken were from the non-reusable category. What is the purpose of studying the non-reusable items?

Dear Reviewer, thank you for your interesting comment. Our initial idea was based on the results of a previous article (PMID: 33805002, doi: 10.3390/dj9040036). In times of pandemic and shortage of PPEs, an attempt was made to devise alternative methods to be able to re-use these PPEs for longer. However, an effective method to disinfect respirators while maintaining the filtering capacity and/or the structure of the mask has not been found yet (Polkinghorne & Branleya, 2020; Fernandes Probst et al. 2021) [References # 4 and 5]. Quite surprisingly, the study in question found that FFP2 masks maintained good efficiency in terms of bacterial filtration efficacy (BFE) even if worn continuously for 40 hours, therefore over several working days (Lines 30-31). In light of these results, we wondered whether the bioburden values too could be able to remain acceptable after prolonged use of these non-reusable devices.

Moreover, also the World Health Organization and the national Centers for Disease Control of many countries have recommended appropriately extending the use time and frequency of single-use respirators (CDC, Centers for Disease Control and Prevention 2020) [Reference #6]. At last, several scientific articles are being published in the literature that reveal how in recent years, probably following the pandemic, the practice of reusing respirators is very common and practiced (Vieira Pereira-Ávila et al. 2020) [Refeence #8]. Reuse and extended use of masks and respirators are very common in many parts of the world, particularly during pandemic and outbreaks and when supplies are short and demand is high (Phin et al. 2009; Chughtai et al. 2015 [References #9 and 10].

We added all these explanations into the text (Lines 36-45).

- Is there any antibacterial or antimicrobial resistance tests conducted on the protective face shields before the real time study with the patient? What is the porosity features of the test masks? What is the age group of the patient?

From this and other comments we realized that we have not been clear enough in writing the methods section. We apologize for this inconvenience. First of all, it is important to specify that the mentioned face shield is a different device from the tested respirators. The shield is a PPE that protects eyes from aerosol; we mentioned it because due to its position, it could have interfered with bacterial contamination of the FFP2 respirators. The shield is a distinct device from the mask, is disinfected after each use and does not come into contact with the respirator. Therefore, it does not alter the state of contamination of the FFP2s. For this reason, we did not test the screen for contamination, since the objective of our preliminary study was to test bioburden on the FFP2 respirators.

In order to avoid further misunderstandings, we added explanatory images (Pages 3 and 4, Figs 1 and 2).

The porosity features of the test masks have been not evaluated by the present research. Asking to the manufactures, we were not able to obtain an exact value for the external layer (the one we have focused on).  The only data we received is that the overall pore diameter was about 30 microns for both.

- Possible microbial loading is the effect primarily due to the surface properties? Is the surface hydrophilic or hydrophobic?

Thank you for your questions. Microbial loading is the result of the deposition on the surface of the mask of the aerosolized products of dental clinical procedures, which are deposited on the surfaces of the mask, of the dental unit, of the gown, etc. Therefore, also the mask surface properties play an important role in the quantity of microbial loading that is found through a test such as that of bioburden. For both models the external layer was of polypropylene that is naturally hydrophobic. We have added this information in the text (Line 85).

This paper does not give any photographic images of the Face shielding masks. It is nice if any representative images are given.

As pointed out in the previous comment, the mentioned face shield is a different device from the tested respirators. The face shield is a PPE that protects eyes from aerosol, and is a distinct device from the mask. As suggested, we added explanatory images (Pages 3 and 4, Figs 1 and 2).

The microstructure of the face shielding masks is necessary. More scientific analysis is required to understand the kind of contaminations. The data present in the study is not sufficient.

Thank you for your comment. As previously mentioned, the face shield is a different device from the tested respirators. The face shield is a PPE that protects eyes from aerosol, and is a distinct device from the mask. To clarify this aspect, we added explanatory images (Pages 3 and 4, Figs 1 and 2).

The kind of contamination we have focused on can be classified as mainly indirect do to the presence of a face shield. Even considering a deeper bacteria characterization an extremely worthy aspect, we have decided to focus only on the general bacterial contamination due to the preliminary study design. This decision was supported by the results of a previous study, where the bacteria on the external surface of surgical masks have been already investigated and characterized for different dental procedures (Gund et al. 2021) [Reference #31].

We also added the following explanation: Considering that both models weight around 9 grams, the external contamination observed after 30 hours of continuative use, remained under the threshold value of 30 CFU/g indicated by the European Law for the respirators production and commercialization (UNI EN 14683:2019). This means that with the present clinical setting both respirators undergo to a limited external contamination even after a long time of use. (Lines 252-257).

Reviewer 2 Report

This manuscript investigated over time the external contamination of two different FFP2s, by studying bioburden values in increasing exposure times. Both respirators were found to be contaminated and this contamination increased with the increase of exposure time. This manuscript can meet the scope of the journal of “Materials”. However, there are still some problems in this manuscript. It needs a major revision before becomes acceptable for publication in this journal.

1. There is only bioburden test for the manuscript. The authors should add other characterizations to support the results of bioburden test.

2. What is the relationship between Results part and Discussion part?

3. There are several language mistakes, and some sentences in the manuscript are confusing or poorly written. For example:

Page 6, “No similar studies have been performed on FFPs, but there is data available regarding surgical masks, unfortunately, however, performed on a total bioburden (internal and external, together.”

Author Response

This manuscript investigated over time the external contamination of two different FFP2s, by studying bioburden values in increasing exposure times. Both respirators were found to be contaminated and this contamination increased with the increase of exposure time. This manuscript can meet the scope of the journal of “Materials”. However, there are still some problems in this manuscript. It needs a major revision before becomes acceptable for publication in this journal.

  1. There is only bioburden test for the manuscript. The authors should add other characterizations to support the results of bioburden test.

Dear reviewer, thank you for your comment. Since the bioburden test makes the tested respirators unusable, we are unable to carry out further analyses on these devices. We added in the text the microstructural description of the two FFP2s analyzed (Lines 85, 204-208).

Moreover, we have added details of the environment in which the experiment took place, the area of the mask in which the bioburden investigation was carried out, and the pre-treatments to which the treated patients were subjected (Lines 102-105, 107-108, 110, 113-114, 119).

  1. What is the relationship between Results part and Discussion part?

Thank you for your question. Since no study of this type exists, it is difficult for us to discuss the results obtained in our article with others previously published in the literature, precisely because there are none. We added this explanation in the text (Lines 217-219). However, we have modified the text by contextualizing the findings of the present study in the Discussion section (Lines 204-208, 216-226, 252-261), and comparing them with the most similar found in the literature [reference #31] (Lines 226-235).

  1. There are several language mistakes, and some sentences in the manuscript are confusing or poorly written. For example: Page 6, “No similar studies have been performed on FFPs, but there is data available regarding surgical masks, unfortunately, however, performed on a total bioburden (internal and external, together.”

Thank you for your comment. We modified the sentence (lines 215-220) and corrected English language mistakes through all the text. All corrections to the manuscript are highlighted in yellow.

Round 2

Reviewer 1 Report

The authors have made serious efforts to make the manuscript with facts and findings. The quality of the manuscript improved after the revision. 

Reviewer 2 Report

Authors answered all the questions. The manuscript is now acceptable for publication in this journal.